# Viral load suppression and HIV-1 drug resistance mutations in persons with HIV on TLD/TAFED in Zambia

**Emmanuel L. Luwaya**[1]*, **Lackson Mwape**[1], **Kaole Bwalya**[1], **Chileleko Siakabanze**[1], **Benson M. Hamooya**[1], **Sepiso K. Masenga**[1,2]

**1** School of Medicine and Health Sciences, Mulungushi University, Livingstone, Zambia, **2** Department of Medicine, Vanderbilt University Medical Center, Nashville, Tennessee, United States of America

* luwayaemmanuel@gmail.com

**Data Availability Statement:** All relevant data are within the paper and its Supporting Information files.

## Abstract

### Background

An increase in the prevalence of HIV drug resistance (HIVDR) has been reported in recent years, especially in persons on non-nucleoside reverse transcriptase inhibitors (NNRTIs) due to their low genetic barrier to mutations. However, there is a paucity of epidemiological data quantifying HIVDR in the era of new drugs like dolutegravir (DTG) in sub-Saharan Africa. We, therefore, sought to determine the prevalence and correlates of viral load (VL) suppression in adult people with HIV (PWH) on a fixed-dose combination of tenofovir diso-proxil fumarate/lamivudine/dolutegravir (TLD) or tenofovir alafenamide/emtricitabine/dolute-gravir (TAFED) and describe patterns of mutations in individuals failing treatment.

### Methods

We conducted a cross-sectional study among 384 adults living with HIV aged ≥15 years between 5th June 2023 and 10th August 2023. Demographic, laboratory and clinical data were collected from electronic health records using a data collection form. Viral load suppression was defined as plasma HIV-1 RNA VL of <1000 copies/ml after being on ART for ≥ 6 months. SPSS version 22 to analyze the data. Descriptive statistics and logistic regression were the statistical methods used.

### Results

The median (interquartile range (IQR)) age was 22 (IQR 18, 38) years, and 66.1% (n = 254) were females. VL suppression was 90.4% (n = 347); (95% confidence interval (CI) 87.6%-93.6%) after switching to TLD/TAFED. Among the virally suppressed, the majority (67.1%, n = 233) were female. Those who missed ≥2 doses in the last 30 days prior to the most recent review were less likely to attain viral suppression compared to those who did not miss any dose (adjusted odds ratio (AOR) 0.047; 95% CI 0.016–0.136; p<0.001). Four participants had resistance mutations to lamivudine and tenofovir. The most common NRTI mutations were M184MV and K65R while K101E was the most common NNRTI mutation.

**Funding:** This work was supported by the Fogarty International Center and National Institute of Diabetes and Digestive and Kidney Diseases of the National Institutes of Health grants 2D43TW009744 (SKM), R21TW012635 (SKM) and the American Heart Association Award Number 24IVPHA1297559 https://doi.org/10.58275/AHA.24IVPHA1297559.pc.gr.193866 (SKM).

**Competing interests:** The authors have declared that no competing interests exist.

**Abbreviations:** ADR, Acquired HIV drug resistance; ART, Antiretroviral therapy; ARV, Antiretroviral (drugs; ATV/r, Atazanavir/ritonavir; CI, Confidence interval; DBS, Dried blood spot; DRM, Drug resistance mutation; DRV/r, Darunavir/ritonavir; DTG, Dolutegravir; GAP, Global Action Plan on HIV drug resistance; HIVDR, HIV drug resistance; LMIC, Low- and middle-income countries; NNRTI, Non-nucleoside reverse-transcriptase inhibitor; NRTI, Nucleoside reverse-transcriptase inhibitor; PDR, Pre-treatment HIV drug resistance; PEPFAR, United States President's Emergency Plan for AIDS Relief; PI, Protease inhibitor; PLWHIV, People living with HIV; TAFED, Tenofovir Alafenamide, Emtricitabine and Dolutegravir; TLD, Tenofovir disproxil fumarate, Lamivudine and Dolutegravir; UNAIDS, Joint United Nations Program on HIV/AIDS; VL, Viral load; WHO, World Health Organization.

## Conclusion

Our findings show that viral suppression was high after switching to TLD/TAFED; but lower than the last 95% target of the UNAIDS. Adherence to antiretroviral therapy was a significant correlate of VL suppression. We, therefore, recommend prompt switching of PWH to TLD/TAFED regimen and close monitoring to enhance adherence to therapy.

## Introduction

The United Nations Program on HIV/AIDS (UNAIDS) has set a fast-track goal of 95-95-95 to end the AIDS pandemic by 2030 [1,2]. The goal is for 95% of people with HIV to be aware of their status, 95% of those who are aware should receive treatment, and 95% of those receiving treatment should be HIV virally suppressed [3]. In Zambia, the latest Zambia Population-based HIV and AIDS Impact Assessment (ZAMPHIA) survey conducted between May-December 2021 reported that 89% of people with HIV knew their status, 98% were receiving treatment, and 96% had viral suppression [4]. While efforts are being made to achieve the target, HIV Drug Resistance (HIVDR) poses a significant threat in the fight against HIV/AIDS [5–8]. There has been an increase in the prevalence of HIVDR in people living with HIV (PLWH), which has been stated as "the emerging threat of HIV drug resistance in Africa" by the World Health Organization (WHO) and UNAIDS [3,9]. Before transition to TLD/TAFED, Zambia reported an estimated 4.3% of individuals on ART for 12–24 months to have some drug resistance [10]. In addition to this, 46.9% of individuals on first-line ART with unsuppressed viral load were reported to have resistance to both Non-Nucleoside Reverse Transcriptase Inhibitors (NNRTIs) and Nucleoside Reverse Transcriptase Inhibitors (NRTIs) [10].

With the increase of HIVDR especially in NNRTIs due to their low genetic barrier to mutations [11–16], WHO recommends the switch to dolutegravir-based first-line regimen in countries where pre-treatment drug resistance to NNRTI reaches the 10% threshold [17]. Dolutegravir (DTG) belongs to the integrase strand transfer inhibitor (INSTI) class of antiretroviral drugs [18]. While drugs in this class are generally highly efficacious, elvitegravir (EVG) and raltegravir (RAL) are susceptible to resistance mutations [1,19–21], leading to virologic failure [22]. DTG however, has shown superiority with regard to genetic barrier to resistance, even more compared to the NNRTI [23]. This has led to recommendation of the use of tenofovir disoproxil fumarate, lamivudine and dolutegravir (TLD) in a single pill, as first- line treatment of HIV regardless of the patient's viral load [23]. Like most sub-Saharan African countries, in 2018, Zambia revised the guidelines for HIV treatment to adopt the recommendations by WHO which emphasized the use of DTG in first-line treatment [24].

Viral load has been used for a long time now as a predictor of response to treatment [25,26]. Individuals with high viral loads ≥1000 copies/microliters are known to be unsuppressed and are recommended for enhanced adherence counselling (EAC) after which another viral load test is done within three months after the last VL test [3]. If still unsuppressed, an HIV drug resistance test is recommended to determine if there are drug resistance mutations [9]. In studies from the Democratic Republic of Congo and Sierra Leone, high viral suppression has been reported in individuals on DTG based-regimen compared to other regimens [27,28]. Although DTG is known to have a high gene barrier to resistance, a recent report from Malawi has shown some resistance after the transition to TLD [29]. Another study in Tanzania also reported resistance DTG [30]. In this study, we sought to determine the

prevalence and correlates of VL suppression and HIVDR mutations in adult PLWH on a fixed-dose combination of TLD or (TAFED) and describe patterns of mutations in individuals failing treatment.

## Methods

### Study design and setting

This study was a retrospective cross-sectional study that was conducted at Arthur Davison Children's Hospital (ADCH) in Ndola district, Zambia, among adults aged ≥15 years, between 5[th], June 2023 and 10[th], August 2023. ADCH antiretroviral clinic is a family center that offers services to people living with HIV regardless of their age in addition to HIV screening and testing. ADCH is also a referral hospital to the central and northern regions of Zambia. At the time of the study, viral load monitoring was routine as guided by the national guidelines.

### Eligibility criteria

The study included adults on TLD/TAFED aged ≥15 years living with HIV. For HIVDR mutation analysis, participants were excluded if they had failed genotyping test despite having a VL >1000 copies/ml, if they had missing data or had been on treatment for less than six months before transition to TLD/TAFED. In addition, HIV RNA sequences were excluded if the treatment regimen at the time of genotyping was unknown. For viral suppression, participants who were on treatment for less than six months before transition to TLD/TAFED were excluded.

### Sample size

Open-source epidemiologic statistics for public health (OpenEpi) online software was used to compute a total sample size of 384 using the maximum estimated frequency of viral load suppression of 50% at a 95% significance level and 80% power in a population of 1000000.

### Data collection and study procedures

Socio-demographic, laboratory and clinical data were collected from medical records using a data collection form. The information was abstracted from electronic-based (SmartCare and sequence files) and paper-based (registers and patient files) health information records. Two sets of viral load data were collected; the first set of viral load data was at least 6 months before participants were switched to TLD/TAFED, while the second set of viral load data was collected after at least 3 months of switching to TLD/TAFED. Data collected in this study was entered into the research electronic data capture (REDCap) before analysis.

We employed a systematic random sampling method to abstract patient files. We determined a sampling interval X by dividing the total number of patient files at a facility by the sample size 384 and taking the whole number only by dropping the decimals. The first file for review was selected (N), by randomly selecting a number between 1 and X (sampling interval). Data abstractors selected the Nth file and then proceeded with selecting file N+X as the first consecutive record for review and continued with adding X until the sample size was been reached. If any patient file from the selected list was excluded from the study, the next eligible record fulfilling the inclusion criteria was selected to replace the excluded file.

### Research variables and definitions

The primary outcome variable of this study was viral load suppression and the secondary outcome variable was drug resistance mutations, while the independent variables were, current

ART regimen, duration on ART, level of education, marital status, age, sex, body mass index (BMI), duration on ART, ART regimen, viral load, number of doses missed in the last 30 days, history of Tuberculosis (TB), history of opportunistic infection, hypertension, diabetic status, full blood count, CD4 count, creatinine and number of sexual partners as indicated in patient file.

Viral load suppression was defined as a VL <1000 copies/microliter of plasma [29], while HIVDR was defined as the presence of at least one drug resistance-associated mutation (DRM) according to the World Health Organization 2009 Surveillance Drug Resistance Mutation list, using the Stanford Calibrated Population Resistance analysis tool (version 4.1 beta, available at http://hivdb.stanford.edu/).

The outcome variable "viral load suppression" was categorized as a binary variable, where viral suppression categories included "virally suppressed <1000 copies/ml" and "unsuppressed viral load >1000 copies/ml". ART regimen before transition to TLD/TAFED were in categories; NNRTI + NRTI, INSTI-based and PI-based regimens. Meanwhile, level of education was defined in four categories; No education, primary, secondary and tertiary education. Variables with binary outcomes such as diabetes, history of tuberculosis, opportunistic infection and viral suppression were defined by yes for the presence or history of the condition or no for the absence of the condition. Hypertension was also defined as such; yes for hypertensive participants and no for non-hypertensive participants.

## Data analysis

Data were imported into SPSS v.22 for analysis. Continuous variables such as age, BMI, hemoglobin and duration on ART were described using medians and interquartile range (IQR). Categorical variables, such as ART regimen, history of TB, viral load suppression, number of people with drug resistance and sex were described using frequency and percentage distributions. To determine the relationship between two categorical variables such viral suppression and ART regimen, a chi-square analysis was performed. Multivariable logistic regression model was used to determine the contribution of each variable toward the outcome and to control for confounding. The multivariate logistic regression included all independent variables with a p-value of < 0.05 at bivalent analysis and those with biological significance. All associations with a P value <0.05 were considered statistically significant. We used strengthening the reporting of observational studies in epidemiology (STROBE) checklist for all sections (S1 File).

## Ethics approval and consent to participate

Ethical approval was obtained from Mulungushi University School of Medicine and Health Sciences Research Ethics Committee (MUHSREC) (Assurance No. FWA0002888 IRB00012281 of IORG0010344 on the 5[th] of April, 2023 and the National Health Research Authority (NHRA) on the 11[th] May, 2023. Permission to conduct research was given by Arthur Davison children's hospital administration. Written consent was waived by the ethics committee due to the retrospective nature of the study where secondary data was used. No personal identifying data were obtained such that it was not possible to identify participants.

## Results

### Characteristic of the study population

The study included a total of 384 participants, with females being the majority, 66.1% (n = 254) and the median age of all participants was 22 years (Interquartile range (IQR), 18, 39). The majority of participants were single (62.8%, n = 241), unemployed (87.2%, n = 337),

attained secondary education (70.1%, n = 269). After switching/initiation on TLD/TAFED, 90.4% participants were virally suppressed (347) compared to 69.7% (267) before switching. Table 1.

## Comparison of socio-demographic and clinical factors between unsuppressed and suppressed participants

A higher proportion of individuals attained viral suppression (90.4%, n = 347). Factors significantly associated with viral suppression included number of doses missed (p<0.001), ART regimen before switching to TLD/TAFED and viral suppression (p<0.001), history of OPI (p<0.001) and Body Mass Index (p = 0.034). Table 2.

## Factors associated with viral suppression in logistic regression

In the univariate analysis, participants with a history of opportunistic infections were 67.7% less likely to attain viral suppression compared to those with a history of no opportunistic infection, odds ratio (OR) 0.323; 95% confidence interval (CI) 0.162, 0.644; p = 0.001. Meanwhile, participants who had missed ≥2 doses in the last 30 days were significantly 95.5% less likely to attain viral suppression compared to those who had not missed any dose in the last 30 days, OR 0.045; 95% CI 0.018, 0.108; p <0.001. Participants who were on a PI-based regimen before switching to TLD/TAFED were significantly 69.4% less likely to attain viral suppression compared to those who were on NNRTI, OR 0.306; 95%CI 0.135, 0.695; p<0.005. Meanwhile, those who were on INSTI-based regimen (AZT/3TC/DTG) before switching to TLD/TAFED, were 79.3% less likely to attain viral suppression compared to those on NNRTI based regimen, OR 0.207; 95%CI 0.074,0.579); p = 0.003, Table 3.

In the multivariate analysis, the factor that was significantly associated with viral load suppression was doses missed in the last 30 days. Compared to those who did not miss a dose in the last 30 days, those who missed ≥2 doses in the last 30 days were 95.3% less likely to attain viral suppression and this was statistically significant (p value <0.001 at 95% CI 0.016,0.136, AOR 0.047); Table 3.

## Distribution of HIV drug (HIVDR) resistance mutations in samples with any HIVDR

Overall, the 3 most common mutations were M184MV (50%), K65R (50%), and K101E (50%). There was only one Major PI mutation M46L (25%) while there were two PI-Accessory mutations V32VALP (25%) and L33LF (25%). The most prevalent mutations were NNRTI mutations, accounting for 9 different types of mutations, while there were 5 different types of NRTI mutations.

## Discussion

The study provides critical data on viral load suppression and drug resistance mutations in adults taking TLD/TAFED at ADCH, Ndola Zambia. In this study we observed an increase in viral load suppression of participants after switching to or initiation on TLD/TAFED regimen. The second observation was that the number of doses missed by participants was a significant factor in viral suppression. Participants who had missed ≥2 doses in the last 30 days were less likely to attain viral suppression compared to those who did not miss a dose in the last 30 days. Finally, we also observed a low prevalence of HIVDR mutations, 0.103% (n = 4), with the pattern of mutations showing NRTI and NNRTI being the most prevalent, 50% each while mutations to PIs and INSTIs were less prevalent and no mutation to dolutegravir were observed.

**Table 1. Study socio-demographic and clinical characteristics.**

| Variables | Frequency/median | Percentage/IQR |
|---|---|---|
| **Age,** *years* | 22 | (18,38) |
| **Sex** | | |
| *Male* | 130 | 33.9 |
| *Female* | 254 | 66.1 |
| **Marital status** | | |
| *Married* | 111 | 28.9 |
| *Single* | 241 | 62.8 |
| *Separated/divorced* | 17 | 4.4 |
| *Widowed* | 15 | 3.9 |
| **Employment status** | | |
| *Formal employment* | 47 | 12.2 |
| *Unemployed* | 337 | 87.2 |
| **Education** | | |
| *No education* | 10 | 2.6 |
| *Primary* | 53 | 13.8 |
| *Secondary* | 269 | 70.1 |
| *Tertiary* | 51 | 13.3 |
| **BMI, kg/m2** | 21 | (19,24) |
| **Hypertension** | | |
| *Yes* | 20 | 5.2 |
| *No* | 363 | 94.5 |
| **Diabetes** | | |
| *Yes* | 1 | 0.3 |
| *No* | 382 | 99.5 |
| **History of TB** | | |
| *Yes* | 24 | 6.3 |
| *No* | 358 | 93.2 |
| **Opportunistic Infection** | | |
| *Yes* | 99 | 25.8 |
| *No* | 284 | 74 |
| **ART Regimen Before** | | |
| *NNRTI + NRTI* | 309 | 80.5 |
| *INSTI-Based* | 23 | 6.0 |
| *PI-Base* | 52 | 13.5 |
| **Viral load Before Switch/Initiation,** *n= 383* | | |
| *1000 copies/ml* | 116 | 30.3 |
| *<1000 copies/ml* | 267 | 69.7 |
| **Viral load After Switch toTLD/TAFED** | | |
| *1000 copies/ml(unsuppressed)* | 37 | 9.6 |
| *<1000 copies/ml(suppressed)* | 347 | 90.4 |
| **Doses missed in past 30 days** | | |
| 0 | 321 | 83.6 |
| 1 | 31 | 8.1 |
| ≥2 | 28 | 7.3 |
| **Duration on ART (months)** | 127 | (95,168) |
| **Hemoglobin** | 12.4 | (11.075,13.3) |

Abbreviations: OPI, opportunistic infection, PI, protease inhibitor, INSTI, integrase strand transfer inhibitor, NNRTI, non-nucleoside reverse transcriptase inhibitor, NRTI, nucleoside reverse transcriptase.

**Table 2. Socio-demographic and clinical factors sorted according to suppressed status.**

| Variables | Viral Load Suppression, n (%) | | P value |
|---|---|---|---|
| | **Yes, 347 (90.4)** | **No, 37 (9.6)** | |
| **Age, (*years*)** | 23 (18,39) | 19 (17,26) | **0.032** |
| **Sex** | | | |
| *Male* | 115 (33.0) | 15 (42.9) | 0.238 |
| *Female* | 233 (67.1) | 21 (57.1) | |
| **Marital status** | | | |
| *Married* | 105 (30.5) | 6 (13.5) | **0.003** |
| *Not Married* | 241 (69.5) | 32 (86.5) | |
| **Employment status** | | | |
| *Formal employment* | 46 (13.3) | 1 (2.7) | 0.066 |
| *Unemployed* | 301 (86.7) | 33 (97.3) | |
| **Education level** | | | |
| *No education* | 8 (2.3) | 2 (5.6) | 0.514 |
| *Primary* | 47 (13.5) | 6 (16.7) | |
| *Secondary* | 244 (70.3) | 25 (67.4) | |
| *Tertiary* | 48 (13.8) | 3 (8.3) | |
| **BMI, kg/m2** | 21 (18,23) | 20 (19,24) | **0.034** |
| **Hypertension** | | | |
| *Yes* | 20 (5.8) | 0 (0) | 0.240 |
| *No* | 328 (94.2) | 35 (100) | |
| **Diabetes** | | | |
| *Yes* | 1 (0.3) | 0 (0) | 1.00 |
| *No* | 347 (99.7) | 35 (100) | |
| **History of TB** | | | |
| *Yes* | 22 (6.4) | 2 (5.4) | 1.00 |
| *No* | 323 (93.6) | 35 (94.6) | |
| **Opportunistic Infection** | | | |
| *Yes* | 81 (23.4) | 18 (48.6) | **0.001** |
| *No* | 265 (76.6) | 19 (51.4) | |
| **ART Regimen** | | | |
| *NNRTI* | 288 (83) | 21 (56.8) | **<0.001** |
| *INSTI-Based* | 17 (4.9) | 6 (16.2) | |
| *PI-Based* | 42 (12.1) | 10 (27) | |
| **Doses missed in past 30 days** | | | |
| 0 | 303 (88.3) | 18 (48.6) | **<0.001** |
| 1 | 28 (8.2) | 3 (8.1) | |
| ≥2 | 12 (3.5) | 16 (43.2) | |
| **Duration on ART (months)** | 125.5 (95,167.3) | 138 (72,192.8) | 0.384 |
| **HB** | 12.4 (11.3,13.3) | 11.4 (10.7,13.2) | 0.188 |

Abbreviations: BMI, body mass index, OPI, opportunistic infection, PI, protease inhibitor, INSTI, integrase strand transfer inhibitor, NNRTI, non-nucleoside reverse transcriptase inhibitor, NRTI, nucleoside reverse transcriptase.

Our study shows that dolutegravir-based regimens are more efficacious and increase level of viral load suppression as it has been reported in other studies [29]. We observed that viral suppression was maintained in most participants, especially those who were previously on NNRTI+NRTI combination such as Tenofovir, Lamivudine, Efavirenz (TLE) and Atripla

**Table 3. Factors associated with viral suppression in logistic regression.**

| Variables | Univariate OR (95%CI) | P value | AOR (95%CI) | P value |
|---|---|---|---|---|
| **Age** | 1.031 (0.999,1.064) | 0.059 | 1.036 (0.986,1.088) | 0.157 |
| **Sex (Male as ref.)** | | | | |
| **Female** | 1.557 (0.783,3.098) | 0.207 | 1.408 (0.570,3.478) | 0.458 |
| **History of OPI (No as ref.)** | | | | |
| **Yes** | 0.323 (0.162,0.644) | **0.001** | 0.556 (0.214,1.446) | 0.229 |
| **Doses missed in the last 30 days (0 as ref.)** | | | | |
| **> or = 2** | 0.045 (0.018,0.108) | **<0.001** | 0.047 (0.016,0.136) | **<0.001** |
| **Regimen before switch/initiation (ref: NNRTI+NRTI)** | | | | |
| **PI-based (TDF/3TC/LPV-r)** | 0.306 (0.135,0.695) | **0.005** | 0.547 (0.191,1.564) | 0.260 |
| **INSTI-based (AZT/3TC/DTG)** | 0.207 (0.074,0.579) | **0.003** | 0.321 (0.090,1.154) | 0.082 |
| **Education level (Ref: No Educ.)** | | | | |
| **Primary** | 1.958 (0.335,11.465) | 0.456 | 2.783 (0.313,24.778) | 0.359 |
| **Secondary** | 2.44 (0.491,12.124) | 0.276 | 5.438 (0.672,44.011) | 0.112 |
| **Tertiary** | 4.00 (0.575,27.819) | 0.161 | 1.520 (0.128,18.097) | 0.741 |
| **Marital Status (Ref: Married)** | | | | |
| **Unmarried** | 0.355 (0.135,0.937) | 0.036 | 0.498 (0.118,2.103) | 0.342 |
| **Employment Status (Ref: Employed** | | | | |
| **Unemployed** | 0.182 (0.024,1.358) | 0.097 | 0.0242 (0.0.16,3.756) | 0.311 |
| **Duration on ART** | 0.999 (0.992,1.006) | 0.766 | 1.004 (0.996,1.012) | 0.369 |

Abbreviations: AZT, Zidovudine, 3TC, lamivudine, LPV-r, liponavir, TDF, tenofovir OPI, opportunistic infection, PI, protease inhibitor, INSTI, integrase strand transfer inhibitor, NNRTI, non-nucleoside reverse transcriptase inhibitor, NRTI, nucleoside reverse transcriptase.

(Tenofovir, Emtricitabine, Efavirenz), and even increased after switching to TLD/TAFED. However, TLD/TAFED have shown superiority in increasing viral suppression and will be key factor in attaining the UNAID target of 95%-95%-95%. This justifies the decision to transition PLWH to TLD/TAFED regardless of their VL. There may be concerns however, that individuals who have high viraemia before switch/initiation to TLD/TAFED are likely to experience viral failure. We observed that some participants with viral failure after switching to TLD/TAFED had a VL >1000 copies/ml before switching and these participants were later reported to have HIV drug resistance mutations after switching to TLD/TAFED. This observation is consistent with other studies where a high viral load before transition is described as a risk for viral failure [25]. Additionally, the high prevalence of HIVDR mutations to NNRTI and NRTI as proven even in other studies, like our study, supports the Zambia consolidated HIV guidelines on the use of dolutegravir which has a high genetic barrier to resistance. This is of more significance because NRTI mutations such as K65R, confer high resistances to tenofovir and lamivudine which are used as backbone for TLD [31]. These mutations were prevalent among individuals failing treatment in our study population. A recent study in Malawi reported treatment-emergent resistance to dolutegravir in two individuals who among few others had viral

load ≥50 copies/m and had NRTI resistance at the time of transition to TLD of which both of them had resistance to both lamivudine and tenofovir. Another study in Tanzania has recently reported HIVDR to dolutegravir in individuals failing treatment. The study reported that three participants with INSTI resistance mutations were highly resistant to dolutegravir and accumulated nucleoside and non-nucleoside RT inhibitor HIVDR mutations [30]. Although our study did not include genetic resistance testing before switching to TLD/TAFED, it is worth noting that all the four individuals that had HIVDR mutations had a VL ≥1000 copies/ ml before switching to TLD/TAFED regimen.

The TLD was recommended by WHO as preferred first-line ART regimen for adolescents and adults due to high genetic barrier to resistance that dolutegravir confers [23] and our findings support the existing evidence of viral suppression attained on TLD with the use of lamivudine and tenofovir disoproxil fumarate as backbone, as compared to NNRTI-based regimen. TAFED is another preferred combination that is commonly used in adults. While drug potency is vital, poor adherence is a fundamental factor to treatment failure in most cases. Our findings show correlation between the number of doses missed and viral suppression, which is in agreement with the ADVANCE and NAMSAL studies in which it was determined that adherence contributes to viral suppression [32,33]. With adherence being a significant factor in viral suppression, there is need to re-evaluate the adherence monitoring methods. The current method applied in our set up at ADCH is based on client disclosure (self-reporting) during the review, which is less objective compared to measuring drug concentration in plasma or urine that can provide more accurate adherence information. With self-reporting, healthcare personnel have to rely on the information given by the client to determine whether or not they have been adherent which makes self-reporting an unreliable method of monitoring adherence. To help manage clients with regard to adherence, review of all those that report to not have missed any dose follow the regular pharmacy schedule of 3 months while those that report to have missed one dose are scheduled for review monthly. Individuals that reported to have missed two or more doses are scheduled for 4 weeks of weekly appointment visits to ensure that they are adhering to treatment. While this may be helpful in some instances still falls short of proper monitoring of adherence. As emphasized in other studies, we reemphasize the need of coming up with proper ways of monitoring treatment adherence of PWLH as this has proven to be an important factor in viral suppression and HIV drug resistance mutations [34].

## Study limitations and strengths

Our study did not include genetic resistance testing before switching to TLD/TAFED. Because of that, we could not establish whether the mutations in the participants were present before switch or developed after switching to the TLD/TAFED regimen. Our study was also limited to one study center, and therefore might not give an accurate picture of the prevalence of viral load suppression and especially HIV-1 drug resistance mutations in Zambia.

Our findings support government's effort to transition adult PLWH to TLD/TAFED in order to attain viral suppression and also gives an understanding of the factors associated with viral suppression. With our large sample size, the findings are a good representation of the study population.

## Conclusions

The study showed an increase in the prevalence of HIV viral suppression after switch from other regimens to dolutegravir based regimen, TLD/TAFED. The number of doses missed by a patient and the regimen before switching correlated with viral suppression. Furthermore, we

observed a low prevalence of HIV drug resistance in participants after 6 months of switching to TLD/TAFED. While there were no drug resistance mutations to dolutegravir observed, NNRTI and NRTI mutations which confer resistance to tenofovir and lamivudine were observed among the few individuals that had HIVDR mutations. Based on our findings, we recommend the re-evaluation of adherence monitoring methods in order to improve patient management and enhance viral suppression.

## Supporting information

**S1 File. Strobe checklist.**
(PDF)

**S2 File. Minimal dataset.**
(XLSX)

**S3 File. Sequence File 1.**
(TXT)

**S4 File. Sequence File 2.**
(TXT)

**S5 File. Sequence File 3.**
(TXT)

## Acknowledgments

We are very grateful to all Laboratory personnel and the Senior Medical Superintendent's office at Arthur Davison Children's Hospital (ADCH) for their support. We would also like to thank the ADCH ART clinic staff for their continued support and assistance during the data collection process.

## Author Contributions

**Conceptualization:** Emmanuel L. Luwaya.

**Data curation:** Emmanuel L. Luwaya, Sepiso K. Masenga.

**Formal analysis:** Sepiso K. Masenga.

**Investigation:** Emmanuel L. Luwaya.

**Methodology:** Emmanuel L. Luwaya.

**Supervision:** Benson M. Hamooya, Sepiso K. Masenga.

**Validation:** Lackson Mwape, Kaole Bwalya, Chileleko Siakabanze, Benson M. Hamooya, Sepiso K. Masenga.

**Visualization:** Lackson Mwape, Kaole Bwalya, Chileleko Siakabanze, Benson M. Hamooya, Sepiso K. Masenga.

**Writing – original draft:** Emmanuel L. Luwaya.

**Writing – review & editing:** Lackson Mwape, Kaole Bwalya, Chileleko Siakabanze, Benson M. Hamooya, Sepiso K. Masenga.

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
