## [Decision Letter · Decision Letter 0]

7 Jun 2024

PONE-D-24-12439Viral load suppression and HIV-1 drug resistance mutations in persons with HIV on TLD/TAFED in Zambia: A cross-sectional studyPLOS ONE

Dear Dr. Luwaya,

Thank you for submitting your manuscript to PLOS ONE. After careful consideration, we feel that it has merit but does not fully meet PLOS ONE’s publication criteria as it currently stands. Therefore, we invite you to submit a revised version of the manuscript that addresses the points raised during the review process.

We look forward to receiving your revised manuscript.

Kind regards,

Anne Kapaata

Academic Editor

PLOS ONE

Journal Requirements:

2. We note that your Data Availability Statement is currently as follows: All relevant data are within the manuscript and its supporting information files.

**Additional Editor Comments:**

Viral load suppression and HIV-1 drug resistance mutations in persons with HIV on TLD/TAFED in Zambia: A cross-sectional study

Review comments

In the background

1. Its mentioned that “In 2021, 75% of people with HIV knew their status, 85% were receiving treatment, and 68% had viral suppression according to a review by Frescura” The precentages seem low, is this the current situation in Zambia as of 2023 when the study was carried out?

2. Provide more literature on levels of HIVDR in the DTG era apart from Malawi…what is the situation in other countries especially Zambia? Are there no other studies that have reported resistance to DTG in zambia? When was DTG adopted in Zambia?

Under methods

1. Correct the dates of carrying out the study is it between 5th, June 2023 and 10th, August 2023 or between May 2023 and August 2023 “

2. For sample size calculation, what guided a viral load suppression of 50%? Is the national rate?

3. What guided the virally suppressed categories of 20 -1000 copies/ml, less 20 copies/ml and Target Not Detected (TND).

Under results

1. What drugs made up the NNRTI +NRTI regimen?

2. Were all the 384 participants non suppressed?

3. How were the OR for viral suprresssion after swutching to DTG calculated?- it appears that these were calculated as if all participants were non-suppressed before switching, therefore giving the impression that PI-based regimen were 85% less likely to suppress after switching, something which is not consistent with the body of literature in this area. See the numbers of the different regimens in table one and two.

4. Table is redundant since number of those with HIVDR mutation is just 4 and its distribution is described in the section before the table

Reviewers' comments:

Reviewer's Responses to Questions

**Comments to the Author**

1. Is the manuscript technically sound, and do the data support the conclusions?

Reviewer #1: Yes

2. Has the statistical analysis been performed appropriately and rigorously? 

Reviewer #1: Yes

3. Have the authors made all data underlying the findings in their manuscript fully available?

Reviewer #1: Yes

4. Is the manuscript presented in an intelligible fashion and written in standard English?

Reviewer #1: Yes

5. Review Comments to the Author

Reviewer #1: The manuscript is interesting and worth publishing. tTe author brought out the science clearly

it is clear that introduction of integrate inhibitors reduces virology failure . a good recommendation for the public health of Zambia

6. PLOS authors have the option to publish the peer review history of their article (what does this mean?). If published, this will include your full peer review and any attached files.

Reviewer #1: No

---

## [Author Response · Author response to Decision Letter 0]

22 Jul 2024

Mulungushi University,

P.O Box 60009,

Livingstone.

22nd July 2024.

The Reviewers,

PLOS ONE Journal.

Dear PLOS ONE reviewers,

RE: RESPONSE TO REVIEWER COMMENTS

Thank you so much for reviewing our manuscript and providing invaluable reviews. We appreciate your review comments which we believe have improved our manuscript's readability and credibility.

We have taken time to correct our manuscript and provided a manuscript copy with track changes to reflect our work on the manuscript.

Below are our responses to the review comments.

Response from authors: Thank you. We have referred to the comments and have since adjusted the naming of the manuscript and supporting documents to PLOS ONE’s style requirements.

2. We note that your Data Availability Statement is currently as follows: All relevant data are within the manuscript and its supporting information files.

Response from authors: Thank you, we have since attached the raw data to this submission.

Response from authors: Thank you, we have since attached the raw data to this submission.

Response from authors: Thank you very much. The title on the online submission has been edited to make identical to the title in the manuscript.

Response from authors: Thank you, we have since deleted the ethics statement in other parts of the manuscript and only retained it in the methods section.

Response from authors: Thank you, we have since made the captions as per recommendation.

Response from authors: Thank you, we have reviewed our reference list as recommended. We have not cited any retracted article.

Review Comments to the Author

Reviewer 1

1. It’s mentioned that “In 2021, 75% of people with HIV knew their status, 85% were receiving treatment, and 68% had viral suppression according to a review by Frescura” The percentages seem low, is this the current situation in Zambia as of 2023 when the study was carried out?

Response from authors: Thank you very much. We have made the revisions and included the correct percentages from the Zambia Population-based HIV and AIDS Impact Assessment (ZAMPHIA) survey conducted between May-December 2021. This is the latest country-wide data showing that Zambia has improved. Among adults (15 years and older) in Zambia living with HIV, 88.7 percent were aware of their status, 98.0 percent of those who knew their status were on ART, and 96.3 percent of those on ART achieved viral load suppression..

2. Provide more literature on levels of HIVDR in the DTG era apart from Malawi…what is the situation in other countries especially Zambia? Are there no other studies that have reported resistance to DTG in Zambia? When was DTG adopted in Zambia?

Response from authors: Thank you so much. We have revised and included the period of adoption of DTG. We’ve also included literature from other countries on DTG. However, according to our knowledge, there are no studies that have reported DTG resistance in Zambia.

Reviewer 2

1. Correct the dates of carrying out the study is it between 5th, June 2023 and 10th, August 2023 or between May 2023 and August 2023 “

Response from author: Thank you for the comment. We have corrected the dates of carrying out of the study, which is between 5th June, 2023 and 10th August, 2023

2. For sample size calculation, what guided a viral load suppression of 50%? Is the national rate?

Response from authors: Thank you so much for the comment. The 50% was used to help obtain the maximal sample size from our study population since national rate was not well documented

3. What guided the virally suppressed categories of 20 -1000 copies/ml, less 20 copies/ml and Target Not Detected (TND).

Response from authors: Thank you very much. The categories were guided by the result output of the viral load instrument that was used. The machine gives result such Target not detected (TND) and less than 20 copies/ml. So the categorization was only for descriptive purposes that provides more data around those that were virally suppressed. However, since the outcome variable “viral load suppression” was categorized as a binary variable, where viral suppression categories included “virally suppressed <1000 copies/ml” and “unsuppressed viral load >1000 copies/ml”, we have now used these categories in the manuscript for consistency.

Under results

1. What drugs made up the NNRTI +NRTI regimen?

Response from author: Thank you so much. The NNRTI + NRTI combination was made up of Tenofovir + Efavirenz + Lamivudine/Emtricitabine. We have since revised and included this description in our results.

2. Were all the 384 participants non-suppressed?

Response from authors: Thank you so much. Not all participants were non-suppressed. Only 116 participants were non-suppressed before switching to DTG and only 35 were non-suppressed after switching to DTG

3. How were the OR for viral suppression after switching to DTG calculated?- it appears that these were calculated as if all participants were non-suppressed before switching, therefore giving the impression that PI-based regimen were 85% less likely to suppress after switching, something which is not consistent with the body of literature in this area. See the numbers of the different regimens in table one and two.

Response from authors:

The OR for the Regimen before switching to DTG-based regimen was calculated for all participants regardless of viral suppression status before switching to DTG. This is because some participants who were virally suppressed before switching became unsuppressed after at last 6 months on DTG suggesting an interplay of other factors contributing to viral suppression. Therefore, to account for changes occurring even in those who were suppressed during the period on DTG, we included them in the analysis.

We have recalculated the OR only for those who were unsuppressed (as shown below), and the multivariable results are not very different in terms of statistical significance. However, neither of the specific regimen were associated with viral suppression in the multivariable model either because the sample size was not highly powered for subgroup analysis or because of confounding factors that have been included in the multivariable regression model. PI based regimen were not significantly associated with viral suppression even on univariable analysis with this change.

Variables Univariate OR (95%CI) P value AOR (95%CI) P value

Regimen before switch/initiation (ref: NNRTI+NRTI) 

PI-based

(TDF/3TC/LPV-r) 0.405 (0.130,1.262) 0.119 0.777 (0.193, 3.135) 0.723

INSTI-based

(AZT/3TC/DTG) 0.135 (0.029,0.627) 0.011 0.350 (0.050, 2.476) 0.293

Note: other variables in the model are not shown here

4. Table is redundant since number of those with HIVDR mutation is just 4 and its distribution is described in the section before the table

Response from authors: Thank you very much for the review. We have revised and removed the table.

We want to thank the reviewers again for taking time to review and make suggestions that has improved our manuscript. We now hope it is acceptable for publication.

Yours,

Emmanuel Luwaya

On behalf of the authors

---

## [Editor Report · Decision Letter 1]

1 Aug 2024

Viral load suppression and HIV-1 drug resistance mutations in persons with HIV on TLD/TAFED in Zambia

PONE-D-24-12439R1

Dear Dr. Luwaya,

We’re pleased to inform you that your manuscript has been judged scientifically suitable for publication and will be formally accepted for publication once it meets all outstanding technical requirements.

Kind regards,

Anne Kapaata

Academic Editor

PLOS ONE

Additional Editor Comments (optional):

The comments have been addressed and the manuscript is ready for publication however all sequences generated as part of this study should be deposited into Genbank.
---

## [Editor Report · Acceptance letter]

29 Aug 2024

PONE-D-24-12439R1 

PLOS ONE

Dear Dr. Luwaya, 

I'm pleased to inform you that your manuscript has been deemed suitable for publication in PLOS ONE. Congratulations! Your manuscript is now being handed over to our production team.

Kind regards, 

on behalf of

Dr. Anne Kapaata 

Academic Editor

PLOS ONE